# Life Quality in Patients with Impaired Visual Acuity Undergoing Intravitreal Medication Applications

**DOI:** 10.3390/ijerph20042879

**Published:** 2023-02-07

**Authors:** Štefanija Kolačko, Jurica Predović, Anamaria Tomić, Valentina Oršulić

**Affiliations:** 1University Hospital “Sveti Duh”, Sveti Duh 64, 10000 Zagreb, Croatia; 2Faculty of Dental Medicine and Health Osijek, Josip Juraj Strossmayer University of Osijek, 31000 Osijek, Croatia; 3Faculty of Medicine, Josip Juraj Strossmayer University of Osijek, 31000 Osijek, Croatia

**Keywords:** patient, intravitreal injections, quality of life, impairment of visual acuity

## Abstract

This cross-sectional study aims to examine the quality of life and difficulties in the daily functioning of patients with impaired visual acuity treated with intravitreal drugs. The survey included 180 adult respondents (78 male and 102 female). The standardized, validated questionnaire VFQ 25 version 2000 was used to measure the quality of life. Results show that, in general, regarding visual functioning, men are significantly more satisfied than women, they rate less intensity of pain, and their distance vision is better. Men report fewer restrictions than women, better color, peripheral vision, and overall visual functioning. The best vision results are in individuals under the age of 60 who also report significantly better social functioning, mental health, fewer restrictions, and less dependence on others. The only significant association between the number of drug applications and the scale of visual functioning is driving motor vehicles—the more applications of the drug they received, the less likely they are to drive a car. The quality of life in patients with chronic ophthalmic diseases treated with intravitreal drugs is reduced, particularly in elderly and female patients who have poorer visual acuity, poorer health in general, and limited social roles.

## 1. Introduction

Quality of life has become an important component that is increasingly used as a subject of research [1]. To date, there are many definitions of quality of life. Felce and Perry define the quality of life as an overall, general well-being that includes objective and subjective factors, as well as the degree of value of material, social and emotional well-being, including personal development, and is all together evaluated through the peculiarities or values of the individual [2,3].

Damage of visual acuity in patients with chronic ophthalmological diseases causes difficulties in everyday functioning and, thus, significantly diminishes their quality of life. Factors favoring the onset of chronic ophthalmic diseases are older age, genetic predispositions, the presence of cataract, obesity, diabetes mellitus, arterial hypertension, hypercholesterolemia, and various comorbidities, and the emphasis is on the female sex [4].

Damage of the macula is one of the leading causes of an irreversible reduction in visual acuity in people over 50 years of age. The macula is responsible for fine central vision, and its damage causes difficulties in reading, recognizing faces, driving a car, and disturbances in color vision [4,5]. When there is a loss of the ability to read, as well as reduced face perception and driving a personal vehicle, patients fall into a difficult emotional state, and normal social functioning becomes difficult, which leads to social isolation. At the same time, the costs of medical and social care for such persons are significant. It is important to recognize difficulties at the earliest stage and prevent the disease progression with adequate therapeutic procedures [6,7].

Intravitreal drug applications are a way of treating various diseases of the retina and choroid, such as the wet form of senile macular degeneration, diabetic macular edema, cystoid macular edema, degenerative myopia, proliferative diabetic retinopathy, occlusion of the central retinal vein and its branches, and other conditions [8,9]. In patients with previously mentioned ophthalmic diseases, there is a significant possibility of a decrease in visual acuity and loss of central vision, which significantly reduces their quality of life, creates difficulties in everyday life and often requires the help of another person. Now, vascular endothelial growth factor (VEGF) antagonists and corticosteroids are the most used intravitreal medications.

The use of intravitreal therapeutics is becoming more frequent in clinical practice and of increasing importance in ophthalmology because it achieves the maximum concentration of the therapeutic substance in the eye, while simultaneously minimizing the possibility of systemic adverse events. The treatment process is being increasingly improved and is becoming the subject of many scientific studies [10,11]. In selected cases, the application of intravitreal therapeutics in the eye stops the progression of the disease, may improve visual function or at least stop further impairment of visual function [12,13].

There is very little research on the quality of life of people with visual acuity impairment; most of them are studies regarding visual acuity and the anatomical results of treatment. In European research, there is little relevant data on the incidence, prevalence, and cause of visual acuity impairment. In Croatia, a study was conducted in 2016 on subjects who had reduced visual acuity [14]. The research aimed to examine how blind and partially sighted people evaluate their quality of life according to the degree of visual impairment, its duration, and participation in rehabilitation. The research showed that the subjective quality of life in people with visual acuity impairment corresponds to the theoretically expected standards; that is, there is a difference in the quality of life according to the degree of impairment. Visually impaired people had a better quality of life than blind people [14].

The 25-item National Eye Institute Visual Function Questionnaire (VFQ-25) is a vision-related quality of life (VR-QOL) instrument designed to assess the patient’s perception of their visual function and quality of life [15].

The primary objective of this study is to measure the quality of life in patients undergoing intravitreal medication applications and to detect age and sex differences between patients.

## 2. Materials and Methods

The research was conducted after obtaining the consent of the Ethical Committee of the University Hospital “Sveti Duh”. The subjects in this research were consecutive patients who came for intravitreal applications of medicine in the eye at the Clinic for Eye Diseases of the University Hospital “Sveti Duh”. Respondents who, due to earlier damage to their visual acuity, were not able to fill out the questionnaire independently and refused the help of a medical professional or refused to participate in the survey were not included in the research. The survey included 180 subjects over the age of 18, of whom 78 were men and 102 were women. No patient was excluded afterwards.

Participation in the research was voluntary and was based on a clear understanding of the research objectives, procedures for implementation, and possible benefits and risks. All of the subjects involved in this research signed informed consent and filled out the standardized questionnaire, VFQ 25 version 2000, which was translated and validated into the Croatian language (previously used in Randomized Study for Efficacy and Safety of Ranibizumab 0.5 mg in Treat-extend and Monthly Regimens in Patients With nAMD (TREND), ClinicalTrials.gov Identifier: NCT01948830). The questionnaire measures the impact of low vision and impaired visual acuity on an individual’s emotional well-being and social functioning; that is, their functioning in everyday life. The questionnaire consists of a basic set of 25 questions, of which 11 are focused on visual acuity, and the rest are general questions focused on the assessment of one’s state of health. The questions were of a closed type [15]. For the collection of the socio-demographic data, a general questionnaire of five questions was created to carry out the mentioned research. One question was open-ended, and four questions were closed-ended. The criterion for the inclusion of the subjects was the arrival for the intravitreal application of the therapeutic drug in the eye via pars plana. 

The research was conducted during June and July 2022. During the research, none of the respondents were excluded. 

## 3. Statistical Methods

The categorical data are represented by absolute and relative frequencies. The numerical data are described by the median and the limits of the interquartile range. The normality of the distribution of the numerical variables was tested by the Shapiro-Wilk test. Differences in the numerical variables between two independent groups were tested by the Mann-Whitney U test, and in the case of three or more groups, by the Kruskal-Wallis test. The association was assessed by Spearman’s correlation coefficient Rho. All P values are two-sided. The significance level was set at Alpha = 0.05. The statistical program MedCalc^®^ Statistical Software version 20.111 (MedCalc Software Ltd., Ostend, Belgium; https://www.medcalc.org, accessed on 24 August 2022) was used for statistical analysis.

## 4. Results

### 4.1. Basic Characteristics of the Respondents

The survey included 180 respondents, of whom 78 (43.3%) were male and 102 (56.7%) were female. Most of the respondents, 111 of them (61.7%), had secondary vocational education. The most common indication for intravitreal drug application was the wet form of age-related macular degeneration (51 cases (28.3%)) and diabetic macular edema (47 cases (26.1%)), while other indications were less common (Table 1).

The median age of the respondents was 70 years, ranging between 28 and 89 years of age. The median number of intravitreal drug applications was 8, ranging between 1 and 41 (Table 2).

### 4.2. Visual Function Assessment

Visual function was assessed by a questionnaire of 25 questions, which formed twelve domains. The theoretical range of the scale was between 0 and 100, where 100 stands for best functioning and 0 for worst functioning.

The median total visual functioning score was 75.5 (interquartile range 59.7 to 85.7). The higher scores were in the domain of eye pain, social functioning, dependence on others, color, and peripheral vision (Table 3).

Regarding visual functioning, the males were significantly more satisfied, compared to the female respondents, with health in general (Mann-Whitney U test, *p* = 0.02), vision in general (Mann-Whitney U test, *p* = 0.01), and the intensity of pain in their eyes being lower (Mann-Whitney U test, *p* = 0.004), and they had better distance vision (Mann-Whitney U test, *p* = 0.003).

In addition, in the group of male subjects, compared to the female subjects, there were fewer limitations (Mann-Whitney U test, *p* = 0.04), and better color vision (Mann-Whitney U test, *p* = 0.03), peripheral vision (Mann-Whitney U test, *p* = 0.03) and overall visual functioning (Mann-Whitney U test, *p* = 0.003) (Table 4).

The best vision in general (Kruskal-Wallis test, *p* < 0.001) and distance vision (Kruskal-Wallis test, *p* < 0.001) were experienced by subjects younger than 60 years of age. Similarly, the respondents under the age of 60 had significantly better social functioning (Kruskal-Wallis test, *p* < 0.001), mental health (Kruskal-Wallis test, *p* = 0.03), fewer limitations (Kruskal-Wallis test, *p* < 0.001), less dependency on others (Kruskal-Wallis test, *p* < 0.001) and in general better visual functioning compared to the respondents who were older than 60 years of age (Kruskal-Wallis test, *p* < 0.001) (Table 5).

The Spearman’s correlation coefficient was used to assess the relationship between the number of drug applications and the scale of visual functioning, and the only significant connection was with driving motor vehicles (Rho = −0.261, *p* = 0.02); that is, the more drug applications the subjects have, the less able they were to drive motor vehicles. vehicles. No significant correlation was shown between the number of drug applications and the other domains (Table 6).

## 5. Discussion

The aim of this work was to examine the quality of life in patients with chronic ophthalmological diseases with impairment of visual acuity and the resulting difficulties in the patient’s daily functioning, who are treated with intravitreal applications of drugs.

Regarding visual functioning, this study showed that the male subjects rated their health condition, color and peripheral vision, and overall visual functioning significantly better than the female subjects. In addition, the men reported less intensity of pain in the eyes, better distance vision, and fewer limitations, they rated social functioning and mental health better, they have fewer limitations in social roles, and are less dependent on others.

Genetic predisposing factors are the subject of numerous studies today. In addition to male-female differences in metabolism, lifestyle, and physical performance, this vast field of research has identified gender differences in response to therapeutic agents, diagnostic, and therapeutic interventions [16]. Furthermore, in the study by Okamoto et al., no differences in age and gender were observed between the groups [17].

The study by the author Korobelnik and associates showed significant functional and anatomical advantages of treatment with intravitreal drug applications in the eye. The endpoint of efficacy was patient who gained 15 or more letters in best-corrected visual acuity at week 24 of treatment, and the improvement achieved was maintained until week 52 of treatment [17].

The previously mentioned data show that the application of the drug in the eye improves visual acuity and prevents the further progression of chronic ophthalmic diseases.

Analyzing the data in the study we conducted in Zagreb, the subjects under the age of 60 rated their visual functioning higher than the older subjects. The best vision in general, and vision at a distance, was experienced by subjects up to the age of 60. It is interesting that the total visual functioning results were better in the >81 years of age group than in the 71–80 years of age group (Table 5). This is probably only a subjective feeling as people older than 81 were probably not burdened with life problems anymore compared to the younger age groups.

Spearman’s correlation coefficient was used to assess the relationship between the number of drug applications in the eye and the scale of visual functioning. There was a significant association between the number of drug applications in the eye and driving a motor vehicle (Rho = −0.261, *p* = 0.02). The more applications of the drug in the eye the patients received, the less able the patients were to drive a motor vehicle. The number of injections in the eye is related to the duration of the disease, poorer response to therapy, and the age of the patient. Therefore, it is understandable that the number of injections is inversely proportional to the ability to drive vehicles.

In an article by Bertelman et al., published in Germany in 2016, the aim was to examine baseline visual function using the VFQ 25 version 2000 questionnaire to assess whether the results differ from the previous phases of the clinical trials. The domains of drug indication, age, gender and their impact on quality of life, and the ability to drive a personal vehicle were examined. Research has shown that the general state of health is affected by the diagnosis of neovascular senile degeneration [18].

A worse quality of life in terms of social functioning, mental health, and limitations in social roles was shown in Bertelman’s article in patients ≥75 years of age, particularly in female subjects, as well as in the case of the research we conducted in Zagreb. The data analysis showed that 10% of respondents gave up driving a motor vehicle due to a decrease in visual acuity and that there is also a tendency for further visual acuity decline with age [18,19].

In the article published by Yuzawa et al., the purpose was to search the available literature and examine the impact of macular degeneration on the patient’s quality of life. By searching scientific databases, they identified all of the articles on quality of life in patients with macular degeneration and its impact on the effect of treatment and quality of life. It was concluded that senile macular degeneration affects the quality of life in the same way as other chronic and systemic diseases, for example, cancer and various heart diseases. They also found that a poorer quality of life affects the daily functioning of patients and results in a higher rate of mental illnesses and a higher risk for early treatment in institutions for the elderly and infirm [20].

In a study by Mangion et al., who investigated the impact of aging on visual functioning, the results show that age-related maculopathy is the cause of worse results on the daily vision activity scale, which consequently causes a worse quality of life [21].

The reduction or loss of visual acuity in an individual leads to changes in social functioning, mental health, and social roles, which consequently reduces their quality of life. It is necessary to encourage patients to respond to ophthalmological examinations as early as possible at the appearance of the first symptoms of the disease to improve visual acuity or reduce its progressive damage with appropriate diagnostic and therapeutic procedures [22,23,24].

Knowing the impact of various factors that affect the quality of life helps optimize patient care and direct the need for medical and mental support provided by the entire healthcare team, to improve patient cooperation and improve quality of life.

The limitations of the study are the relatively small number of subjects, the lack of a control group, and the lack of a comparison with visual acuity, which opens opportunities for further research work.

## 6. Conclusions

Based on the conducted research and obtained results, we came to the following conclusions:The quality of life in patients with chronic ophthalmological diseases is reduced, particularly in subjects over 60 years of age. They had poorer health in general, greater impairment of visual acuity, and limitations in social roles, which is expected due to the natural process of aging.Regarding visual functioning, the male subjects were significantly more satisfied than the female subjects. In addition, compared to the female subjects, the male subjects rated their health in general, vision in general, and vision at a distance better and reported less intensity of pain in the eyes.By assessing the relationship between the number of intravitreal drug applications and the scale of visual functioning, the only significant association was between the number of intravitreal drug applications and driving a motor vehicle. The results show that the more applications the patient had, the less able they were to drive a motor vehicle. The application of more injections, the consequence of which is the greatest impediment to driving vehicles, might also be due to a greater progression of the disease or a lack of response to treatment.It is necessary to encourage patients to respond to ophthalmological examinations as early as possible, at the appearance of the first symptoms of the disease, to improve visual acuity or reduce its progressive damage with appropriate diagnostic and therapeutic procedures.

## Figures and Tables

**Table 1 ijerph-20-02879-t001:** Basic characteristics of the respondents.

	No (%)
Gender	
Male	78 (43.3)
Female	102 (56.7)
Level of education	
Elementary School	21 (11.7)
High school	111 (61.7)
College education	16 (8.9)
Higher vocational education	28 (15.6)
PhD	4 (2.2)
Indication for intravitreal therapy	
Wet form of age-related macular degeneration	51 (28.3)
Diabetic macular edema	47 (26.1)
Retinal vein occlusion	25 (13.9)
Proliferative diabetic retinopathy	14 (7.8)
Choroidal neovascularization	12 (6.7)
Other	31 (17.2)
Total	180 (100)

**Table 2 ijerph-20-02879-t002:** Age of the respondents and ordinal number of current intravitreal therapy application.

	Median (Interquartile Range)	Range
Age of respondents (years)	70 (63–77)	28–89
No of intravitreal therapy application	8 (3–14)	1–41

**Table 3 ijerph-20-02879-t003:** Assessments of visual functioning (VFQ 25 questionnaire, ver. 2000).

	Median (Interquartile Range)	Range
Health in general	50 (25–50)	0–100
Sight in general	60 (40–80)	20–100
Pain in the eyes	100 (75–100)	0–100
Near vision	66.7 (41.7–83.3)	0–100
Distance vision	83.3 (58.3–100)	8.3–100
Vision related score		
Social functioning	100 (75–100)	12.5–100
Well-being, mental health, well-being	71.9 (43.8–87.5)	0–100
Social role limitations	75 (37.5–87.5)	0–100
Dependence on others	100 (66.7–100)	0–100
Management of motor vehicles	83.3 (66.7–100)	0–100
Color vision	100 (100–100)	25–100
Peripheral vision	100 (50–100)	25–100
Total visual functioning	75.5 (59.7–85.7)	11–98

**Table 4 ijerph-20-02879-t004:** Gender differences in visual functioning.

	Median (Interquartile Range)		*p* *
	Men	Women	
Health in general	50 (25–50)	25 (0–50)	**0.02**
Sight in general	60 (40–80)	52 (40–60)	**0.01**
Pain in the eyes	100 (87.5–100)	87.5 (75–100)	**0.004**
Near vision	70.8 (50–83.3)	58.3 (41.7–83.3)	0.09
Distance vision	85.42 (66.7–100)	75 (50–91.67)	**0.003**
Vision related score			
Social functioning	100 (75–100)	87.5 (75–100)	0.09
Well-being. mental health. well-being	75 (50–87.5)	68.8 (31.5–87.5)	0.06
Social role limitations	75 (50–100)	60 (25–87.5)	**0.04**
Dependence on others	100 (75–100)	91.67 (50–100)	0.06
Management of motor vehicles	83.3 (66.7–95.8)	83.3 (58.3–100)	0.81
Color vision	100 (100–100)	90 (75–100)	**0.03**
Peripheral vision	100 (75–100)	75 (50–100)	**0.04**
Total visual functioning	79.05 (64.4–88.7)	72.16 (53.1–84.0)	**0.003**

* Mann-Whitney U test. Bold numbers = statistically significant results.

**Table 5 ijerph-20-02879-t005:** Assessments of visual functioning regarding age groups.

	Median (Interquartile Range)				*p **
	<60 Years	61–70 Years	71–80 Years	>81 Years	
Health in general	50 (25–62.5)	50 (25–50)	25 (6.3–50)	50 (25–50)	0.50
Sight in general	80 (60–80)	60 (45–80)	40 (40–60)	40 (20–60)	**<0.001**
Pain in the eyes	100 (75–100)	100 (75–100)	93.8 (75–100)	100 (75–100)	0.77
Near vision	75 (58.3–83.3)	66.7 (41.7–83.3)	58.3 (41.7–75)	66.7 (50–91.7)	0.15
Distance vision	91.7 (75–100)	83.3 (58.3–100)	66.7 (41.7–89.6)	75 (66.7–91.7)	**<0.001**
Vision related					
Social functioning	100 (100–100)	90 (75–100)	87.5 (62.5–100)	87.5 (37.5–100)	**<0.001**
Well-being, mental health	87.5 (50–93.8)	75 (43.8–87.5)	65.6 (31.3–81.3)	56.3 (25–81.3)	**0.03**
Social role limitations	100 (56.3–100)	75 (50–100)	56.3 (25–75)	75 (25–75)	**<0.001**
Dependence on others	100 (95.8–100)	100 (75–100)	87.5 (41.7–100)	75 (16.7–100)	**<0.001**
Management of motor vehicles	91.7 (75–100)	83.3 (66.7–91.7)	75 (58.3–100)	66 (n = 1)	0.51
Color vision	100 (100–100)	100 (100–100)	100 (75–100)	100 (100–100)	0.07
Peripheral vision	100 (75–100)	87.5 (75–100)	75 (50–100)	100 (50–100)	0.15
Total visual functioning	85 (74–90.2)	77.6 (60.4–85.8)	68.1 (47.3–81.9)	71.6 (59.3–77.9)	**<0.001**

* Kruskal-Wallis test. Bold numbers = statistically significant results.

**Table 6 ijerph-20-02879-t006:** Correlation of visual functioning with the number of drug applications (Spearman’s correlation coefficient).

	Correlation with the Number of Intravitreal Injections (Spearman’s Correlation Coefficient)	
	Rho	*p* Value
Health in general		0.42
Sight in general	0.024	0.74
Pain in the eyes	0.001	>0.99
Near vision	−0.075	0.32
Distance vision	−0.107	0.15
Related to vision		
Social functioning	−0.007	0.92
Well-being, mental health, well-being	−0.072	0.34
Social role limitations	−0.085	0.26
Dependence on others	−0.084	0.26
Management of motor vehicles	**−0.261**	**0.02**
Color vision	−0.006	0.93
Peripheral vision	−0.008	0.92
Total visual functioning	−0.121	0.11

Bold numbers = statistically significant results.

## Data Availability

The data presented in this study are available on request from the corresponding author. The data are not publicly available due to privacy and ethical reasons.

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
