# Peer review of "Life Quality in Patients with Impaired Visual Acuity Undergoing Intravitreal Medication Applications"

_ijerph, 2023, doi:10.3390/ijerph20042879_

Round 1
Reviewer 1 Report
Dear authors,
I consider that the topic of the manuscript is interesting and suitable for publication in IJREPH. The text is well-structured and easy to read. Still, there are some revisions, as follows:
- There are some writing/editing mistakes: i.e. lines 30, 73, 99, etc. (extra spaces); line 38 (a dot before de reference number); lines 66, 112 (double dots); etc.
- Page 1, line 17: “Also, they report significantly” – it is not clear which are “they”, please rephrase.
- Page 2, Materials and method.
o Please detail the selection of the subjects (the method – random?).
o Did any patient refuse?
o What were the exclusion criteria? But the benefits?
o The VFQ 25 questionnaire is validated for Croatian people (linguistic point of view and psychometric properties)? If so, please add the references. Was it pretested?
o Please detail more about the structure/domain (twelve domains) of the questionnaire.
o the control lot is missing.
- Page 4, lines 125-126: what do you mean by “The 125 best results” Please rephrase.
- Page 7: mention the limits of the study.
- Page 7, lines 228-230: the fragment is more appropriate for the conclusion part, instead of conclusion no. 4. Also, I think there is a need also for the health education of patients for regular check-ups (early detection).
- Page 7, conclusion: please add the practical implication for clinicians who read the article.
Author Response
Dear reviewer,
thank You for meticulous and comprehensive review and constructive comments which I incorporated in the attached manuscript.
Yours sincerely,
Jurica Predović

Reviewer 2 Report
I would like to congratulate the authors for the work because it has been interesting and enjoyable to read. However, in the spirit of being constructive, I would like to make some comments that I have collected in the attached document.

Author Response

(The authors gave the same response as above.)

Round 2
Reviewer 1 Report
Dear authors, the manuscript has been improved, and I agree with the publication in the present form.
Regards,
Author Response
Dear reviewer,
thank You for the effort You made during revision of our manuscript and giving approval for the publication.
Kind regards,
Jurica Predović
Reviewer 2 Report
In general, many of the appreciations that I mentioned in my previous review have been taken into account, for which I thank you. However, I cannot find the modification that you say you have made to the second comment that I make to you in the introduction section anywhere in the modified text. On the other hand, in the objectives of the study that have been added at the end of the introduction, there is a mistake in the last sentence. I understand that they mean “sex”.
Author Response
Dear reviewer,
I accepted and implemented Your comments in the article with a joy, Your comments really improved our manuscript, thank You once again for the effort You made to improve our work.
Yours,
Jurica Predović